# FTO Intronic SNP Strongly Influences Human Neck Adipocyte Browning Determined by Tissue and PPARγ Specific Regulation: A Transcriptome Analysis

**DOI:** 10.3390/cells9040987

**Published:** 2020-04-16

**Authors:** Beáta B. Tóth, Rini Arianti, Abhirup Shaw, Attila Vámos, Zoltán Veréb, Szilárd Póliska, Ferenc Győry, Zsolt Bacso, László Fésüs, Endre Kristóf

**Affiliations:** 1Department of Biochemistry and Molecular Biology, Faculty of Medicine, University of Debrecen, H-4032 Debrecen, Hungary; toth.beata@med.unideb.hu (B.B.T.); ariantirini@med.unideb.hu (R.A.); abhirup.shaw@med.unideb.hu (A.S.); vamos.attila@med.unideb.hu (A.V.); poliska@med.unideb.hu (S.P.); 2Doctoral School of Molecular Cell and Immune Biology, University of Debrecen, H-4032 Debrecen, Hungary; 3Regenerative Medicine and Cellular Pharmacology Research Laboratory, Department of Dermatology and Allergology, University of Szeged, H-6720 Szeged, Hungary; 4HCEMM-SZTE Skin Research Group, University of Szeged, H-6720 Szeged, Hungary; 5Department of Surgery, Faculty of Medicine, University of Debrecen, H-4032 Debrecen, Hungary; fgyory@freemail.hu; 6Department of Biophysics and Cell Biology, Faculties of Medicine and Pharmacology, University of Debrecen, H-4032 Debrecen, Hungary; bacso@med.unideb.hu

**Keywords:** adipocyte browning, differential gene expression patterns, deep-neck, PPARγ, FTO obesity-risk allele

## Abstract

Brown adipocytes, abundant in deep-neck (DN) area in humans, are thermogenic with anti-obesity potential. FTO pro-obesity rs1421085 T-to-C single-nucleotide polymorphism (SNP) shifts differentiation program towards white adipocytes in subcutaneous fat. Human adipose-derived stromal cells were obtained from subcutaneous neck (SC) and DN fat of nine donors, of which 3-3 carried risk-free (T/T), heterozygous or obesity-risk (C/C) FTO genotypes. They were differentiated to white and brown (long-term Peroxisome proliferator-activated receptor gamma (PPARγ) stimulation) adipocytes; then, global RNA sequencing was performed and differentially expressed genes (DEGs) were compared. DN and SC progenitors had similar adipocyte differentiation potential but differed in DEGs. DN adipocytes displayed higher browning features according to ProFAT or BATLAS scores and characteristic DEG patterns revealing associated pathways which were highly expressed (thermogenesis, interferon, cytokine, and retinoic acid, with *UCP1* and *BMP4* as prominent network stabilizers) or downregulated (particularly extracellular matrix remodeling) compared to SC ones. Part of DEGs in either DN or SC browning was PPARγ-dependent. Presence of the FTO obesity-risk allele suppressed the expression of mitochondrial and thermogenesis genes with a striking resemblance between affected pathways and those appearing in ProFAT and BATLAS, underlining the importance of metabolic and mitochondrial pathways in thermogenesis. Among overlapping regulatory influences that determine browning and thermogenic potential of neck adipocytes, FTO genetic background has a thus far not recognized prominence.

## 1. Introduction

Brown and beige adipocytes play a major role in maintaining the constant core body temperature of hibernating, small, and newborn animals, as well as in humans without shivering [1,2]. Their heat production is mainly mediated by UCP1, a mitochondrial carrier protein, which uncouples ATP synthesis from the respiratory chain activity [1,3]. These adipocytes also conduct effective UCP1-independent thermogenic mechanisms [4,5,6,7,8,9]. The stimulation of these processes leads to increased energy expenditure that can ameliorate the energy balance during obesity and type 2 diabetes mellitus [10,11].

In rodents, brown adipose tissue (BAT) contains classical brown adipocytes which derive from myogenic precursors, accumulate numerous small lipid droplets in a multilocular arrangement and convert glucose and fatty acids into heat mostly by the action of the constitutively expressed UCP1 [1,2,12]. Beige adipocytes with similar morphologic features were also described in mice. These cells arise from mesenchymal precursors and have a common developmental origin with white adipocytes. They exist in distinct thermogenic fat depots and can be induced by cold and subsequent adrenergic stimulation [2,13]. In adult humans, specific adipose depots are enriched in brown adipocytes; these expanse to 1–1.5% of total body mass and are mostly found in the perirenal, deep-neck (DN), and paravertebral regions [14]. It is still unrevealed whether these thermogenic fat depots represent the classical brown or the beige type of adipocytes by origin and function, even after recent intense studies of DN tissue, which could be compared to paired subcutaneous (SC) fat samples [15,16,17,18]. For simplicity, in this paper, the term brown is used to cover both classical and beige thermogenic adipocytes in humans.

Human brown adipocytes differentiated in distinct adipose tissues express thermogenic genes at moderate levels under unstimulated conditions [2,14]. The ratio of brown and white adipocytes is partially determined during the early differentiation of mesenchymal progenitors into adipocyte subtypes, which is strongly influenced by genetic predisposition [19,20]. This can be quantified in a given tissue or cell culture sample by determining BATLAS score based on the expression of 98 brown and 21 white-specific genes, which were selected by a combined analysis of gene expression signatures in murine brown, beige, and white adipocytes and human tissue samples [21]. Similarly, browning probability can be estimated by a recently developed computational tool, ProFAT [22].

A recent genome-wide association study of body fat distribution identified 98 independent adiposity loci which could affect the appearance of thermogenic fat [23]. In a detailed study by Claussnitzer et al., it has been described that single-nucleotide polymorphism (SNP) rs1421085 underlies the genetic association between fat mass and obesity-associated (FTO) locus; obesity and the presence of the C risk-allele of the FTO locus results in a cell-autonomous, IRX3- and IRX5-dependent shift in the gene expression programs, generating white adipocytes instead of brown with lipid-storage and decreased thermogenesis. When the T/T risk-free genotype is carried at the rs1421085 position, the ARIDB5 repressor becomes able to bind to this site not allowing the expression of IRX3 and IRX5 that let adipocyte precursors to be committed for brown differentiation [19].

To learn more about human adipocyte browning and attempting to through more light on unresolved or controversial issues in the regulation of thermogenesis, we decided to study neck adipocyte populations derived from primary human adipose-derived stromal cells (hASCs) instead of whole tissue samples with different cell types present or single-cell clones influenced by immortalization protocols. We screened and compared global gene expression patterns by RNA sequencing of hASC-derived white and brown (in response to sustained PPARγ stimulation) differentiated adipocytes. The hASCs were isolated from paired DN and SC adipose tissue samples of nine donors, three of each FTO rs1421085 genotype: T/T-risk-free, T/C-heterozygous, and C/C-obesity-risk. We found that both brown and beige markers, including *UCP1*, *CKMT1A/B* [4,15], *CIDEA* [24], and *PM20D1* [6], were upregulated in DN as compared to SC adipocytes, indicating a large potential of thermogenesis in the deep depots. Novel pathways and biological processes could be linked to browning regulation, comparing patterns of upregulated genes and disentangling the complicated set of interactions at protein level which may point out the most indispensable proteins need to maintain the particular phenotype. On the other hand, dozens of genes (such as *CIDEA*, *CITED1* [25], and *PM20D1* thermogenic markers) were upregulated in response to the brown differentiation as compared to white irrespective of the anatomical origin of the hASCs. The gene expression pattern of brown adipocytes was determined to a greater extent by the anatomical origin of the hASCs from which they had been differentiated than the differentiation protocol. Surprisingly, the expression of metabolic, mitochondrial, and thermogenic genes was strikingly compromised by the presence of FTO rs1421085 genotypes in the progenitors. Our results suggest that cultivated hASCs from distinct locations of the human neck still keep their differing propensity for adipocyte browning which is strongly influenced by the presence of the obesity-risk alleles at the FTO intronic locus.

## 2. Materials and Methods

### 2.1. Ethics Statement and Obtained Samples, Isolation, and Differentiation of hASCs

Tissue collection was complied with the guidelines of the Helsinki Declaration and was approved by the Medical Research Council of Hungary (20571-2/2017/EKU) followed by the EU Member States’ Directive 2004/23/EC on presumed consent practice for tissue collection. Written informed consent was obtained from all participants before the surgical procedure. hASCs were isolated from SC and DN adipose tissue of volunteers (BMI < 29.9) aged 35–75 years after written informed consent. During thyroid surgeries, a pair of DN and SC adipose tissue samples was obtained to rule out inter-individual variations. Patients with known diabetes, malignant tumor, or with abnormal thyroid hormone levels at the time of surgery were excluded [26].

hASCs were isolated and cultivated; white and brown adipocytes were differentiated from hASCs according to described protocols [6,7,26,27]. Briefly, both white and brown differentiations were induced by hormonal cocktails in serum and additive-free DMEM-F12-HAM medium that contain apo-transferrin (Sigma-Aldrich, Munich, Germany cat#T2252), insulin (Sigma-Aldrich cat#I9278), T3 (Sigma-Aldrich cat#T6397), dexamethasone (Sigma-Aldrich cat#D1756), and IBMX (Sigma-Aldrich cat#I5879). Later, dexamethasone and IBMX were omitted from both media. In the white cocktail hydrocortisone (Sigma-Aldrich cat#H0888) was constantly present, while the brown contained insulin at 40× higher concentration than the white. However, the major difference between the two protocols was the time interval and concentration of the administered rosiglitazone (Cayman Chemicals, Ann Arbor, MI, USA cat#71740). While the white regimen contained 2 µM rosiglitazone on the first four days of the two weeks long process, the differentiating brown adipocytes were treated with the drug at 500 nM concentration between the fourth and 14th days [28,29]. The absence of mycoplasma was checked by polymerase chain reaction (PCR) analysis (PCR Mycoplasma Test Kit I/C, PromoCell, Heidelberg, Germany cat# PK-CA91) [7,26].

### 2.2. Flow Cytometry

To investigate the phenotype of the undifferentiated hASCs, multiparametric analysis of surface antigen expression was performed by three-color flow cytometry using fluorochrome-conjugated antibodies with isotype matching controls. See [7,30] for further details about the analysis.

### 2.3. RNA and DNA Isolation, Genotyping

RNA and DNA preparation was carried out as described previously [6,7,26,27]. Rs1421085 SNP was genotyped by qPCR using TaqMan SNP Genotyping assay (Thermo Fisher Scientific, Waltham, MA, USA cat#4351379) according to the Manufacturer’s instructions [7].

### 2.4. RNA-Sequencing

To obtain global transcriptome data, high throughput mRNA sequencing analysis was performed on Illumina sequencing platform. Total RNA sample quality was checked on Agilent BioAnalyzer using Eukaryotic Total RNA Nano Kit according to the Manufacturer’s protocol. Samples with RNA integrity number (RIN) value > 7 were accepted for the library preparation process. Libraries were prepared from total RNA using NEBNext^®^ Ultra™ II RNA Library Prep for Illumina (New England BioLabs, Ipswich, MA, USA) according to the manufacturer’s protocol. Briefly, poly-A RNAs were captured by oligo-dT conjugated magnetic beads then the mRNAs were eluted and fragmented at 940 °C. First-strand cDNA was generated by random priming reverse transcription and after second strand synthesis step, double-stranded cDNA was generated. After repairing ends, A-tailing and adapter ligation steps, adapter-ligated fragments were amplified in enrichment PCR and finally sequencing libraries were generated. Sequencing runs were executed on Illumina NextSeq500 instrument using single-end 75 cycles sequencing.

After sequencing, the reads were aligned to the GRCh38 reference genome (with EnsEMBL 95 annotation) using STAR aligner (version 2.7.0a) [31]. To quantify our reads to genes, featureCounts was used (version: 1.6.3) [32].

Subsequent gene expression analyses were performed in R (version 3.5.2). Genes with low expression values or with outlier value were removed from further analysis. Briefly, after removing the highest read value of each gene, we filtered out genes with reads less than 50, considering raw reads count. Then, to further remove outlier genes, we calculated Cook’s distance and filtered out genes with Cook’s distance higher than 1. After filtering out the low-expressed and outlier ones, the expression profile of 22,362 transcripts could be examined. PCA analysis did not show any batch effect, considering sequencing date, the donor origin, and the donor sex or tissue origin. Differential expression analysis was performed using DESeq2 algorithm (version 1.22.2). When tissue origin and differentiation protocol based differential expression was determined donor origin was controlled to decrease the effect of biological variance on thermogenic capacity [33]. However, when the comparison was based on the FTO obesity-risk allele presence, we did not control donor origin. Significantly differentially expressed genes (DEGs) were defined based on adjusted *p* values < 0.05 and log2 fold change threshold > 0.85.

Hierarchical cluster analyses and heat map visualization was performed on the Morpheus web tool (https://software.broadinstitute.org/morpheus/) using Pearson correlation of rows and columns and complete linkage based on calculated z-score of DESeq normalized data after log_2_ transformation. The z-score was calculated in two ways: to eliminate the donor background, we calculated it by donors, and to examining FTO obesity-risk allele presence-based donor differences, we determined it by considering all samples, as indicated in the figures. The interaction network and significantly enriched pathways (Reactome and KEGG or Biological processes) were determined by STRING (https://string-db.org). Edges represent protein-protein based interactions. To identify the Interaction network, we took in consideration text mining, databases, and experiments from the active interaction source from the STRING database; interaction strength score threshold was 0.4. Further analyzes of the STRING network data was performed in igraph R to identify betweenness score and bridges numbers of the interacting nodes. Different interaction score confidence value (0.4–0.7) was used to visualize the network, as indicated in the figures, to obtain the interpretable complexity of the network. Interactomes were also constructed by Gephi, which visualizes the fold change of the genes. Prediction of browning capacity was performed using PROFAT (http://ido.helmholtz-muenchen.de/profat/) and BATLAS (http://green-l-12.ethz.ch:3838//BATLAS/) computational tools.

RNAseq data have been deposited to Sequence Read Archive (SRA) database (https://www.ncbi.nlm.nih.gov/sra) under accession number PRJNA607438.

### 2.5. Antibodies and Immunoblotting

Lysis of differentiated adipocytes, denaturation, SDS-PAGE, and blotting were performed as described previously. Overnight, membranes were probed at 4 °C with primary antibodies: polyclonal anti-UCP1 (1:500, Sigma-Aldrich cat#U6382), and anti-actin (1:10000, Sigma-Aldrich cat#A2066) in TBS-T containing 1% non-fat skimmed milk, followed by the incubation with horseradish-peroxidase-conjugated anti-rabbit secondary antibody (Sigma-Aldrich Cat#A1949) for 1 h at room temperature. Immunoreactive proteins were visualized by Immobilion western chemiluminescence substrate (Merck-Millipore, Darmstadt, Germany cat#WBKLS0500) [7].

### 2.6. Immunofluorescence Staining, Quantification of Browning

hASCs were plated and differentiated on Ibidi eight-well μ-slides; vital and immunofluorescence staining was carried out as described previously [6,7,26,27]. Sample scanning was done by iCys Research Imaging Cytometer (iCys, Thorlabs Imaging Systems, Sterling, VA, USA). The images were processed and analyzed by our high-throughput automatic cell-recognition protocol using the iCys companion software (iNovator Application Development Toolkit version 7.0, CompuCyte Corporation, Westwood, MA, USA), Cell Profiler and Cell Profiler Analyst (The Broad Institute of MIT, MA, USA). Texture sum variance and median Ucp1 protein content of adipocytes (as compared to SC white adipocytes) per cell was determined. In every experiment, 2000 cells per each sample were recorded and measured. See [7,27,34] for further details about the analysis.

### 2.7. Determination of Cellular Oxygen Consumption (OC) and Extracellular Acidification Rate (ECAR)

OC and ECAR were measured using an XF96 oxymeter (Seahorse Biosciences, North Billerica, MA, USA). hASCs of donors not related to the RNA-Seq analysis were seeded and differentiated in 96-well XF96 assay plates. Baseline, dibutyril-cAMP (Sigma-Aldrich cat#D0627) stimulated, β-guanidinopropionic acid (Sigma-Aldrich cat#G6878), and oligomycin (Enzo Life Sciences, Farmingdale, NY, USA cat#ALX-380-037) inhibited OC and ECAR were recorded. As the last step, cells received a single bolus dose of Antimycin A (Sigma-Aldrich cat# U8674) at 10 μM final concentration for baseline correction. The oxygen consumption rate (OCR) and ECAR were normalized to protein content and normalized readings were shown. For statistical analysis, relative OC and ECAR levels were determined to compare basal, cAMP stimulated and oligomycin inhibited (both in unstimulated and stimulated cells) OCRs/ECARs of each sample to the basal OCR/ECAR of untreated SC white adipocytes [6,7,26].

### 2.8. Statistical Analysis

Results are expressed as the mean ± s.d. for the number of donors indicated. The normality of distribution of the data was tested by Shapiro-Wilk test. In a comparison of two groups, two-tailed Student’s t-test was used. For multiple comparisons of groups statistical significance was calculated and evaluated by two-way ANOVA followed by Tukey post-hoc test. Statistical analysis of differential expression was performed in the R programing language with DESeq [35]. Hierarchical cluster analysis during heat map generation was performed based on Pearson correlation. The *n* values represent biological replicates. Specific details for *n* values are noted in each figure legend. 

## 3. Results

### 3.1. DN and SC Progenitors Have Similar Surface Markers and Adipocyte Differentiation Potential But Differ in the Gene Expression Profile

We intended to examine global gene expression patterns of hASC-derived white and brown adipocytes and undifferentiated progenitors cultivated from paired DN and SC adipose tissue biopsies. Before doing that, we aimed to validate if adipocytes differentiated for two weeks from the aforementioned hASCs isolated from four individuals have different functional properties as described in previous studies [18,36]. Basal and proton leak mitochondrial respiration of DN adipocytes was significantly elevated as compared to the SC adipocytes regardless of the obtained protocol (Appendix A). After the cells received a single bolus dose of cell-permeable dibutyril-cAMP, mimicking adrenergic stimulation, we found that adipocytes that were differentiated from DN precursors were more capable to activate their respiration than the adipocytes of SC origin. In parallel, basal and cAMP-stimulated ECARs of DN adipocytes were greater in a significant degree than those of the SC ones (Appendix A). We could also estimate, by applying the creatine analogue β-GPA, the contribution of UCP1-independent creatine futile cycle to oxygen consumption [4] and found it was more pronounced in DN adipocytes (Appendix A).

As a further characterization of the hASCs, surface antigen analysis was performed. There was no significant difference in the presence of hematopoietic/monocyte, fibroblast, endothelial, and integrin cell surface markers between the SC and DN precursors (Figure 1A and Appendix A). Less than 5% of them had CD31, CD45, and CD146 markers, which exclude substantial endothelial or leukocyte contamination. Primary abdominal subcutaneous hASCs and SGBS cells, which can also be differentiated to both white and brown adipocytes [7,27,34] that differed in the presence of the MCAM marker CD146.

Comparing the global gene expression profile of the hASCs obtained from the two anatomical sites of nine donors, large differences were observed. Numbers 1 to 9 on corresponding figures throughout the manuscript label individual donors from whom the samples were obtained. In this comparison, 1420 genes were differentially expressed: 878 were higher in DN and 542 in SC progenitors, respectively (Figure 1B). The list of significantly enriched pathways (only 10 are shown), determined by considering the higher expressed genes, is very different between the two progenitors, except the similarly high expression of genes related to extracellular matrix (ECM) regulation. The DN hASC profile was dominated by the complement and coagulation cascade, retinoic acid biosynthesis and signaling, interaction between L1 and ankyrins, and neuronal systems and hemostasis pathways. SC precursors showed characteristic enrichment for homeobox/homeodomain mediated transcription regulation, neuroactive ligand-receptor interaction, PPAR signaling, synthesis of GPI-anchor proteins, and integrin cell surface interaction pathways (Figure 1C–F; Italic fonts indicate enriched pathway, biological process or compounds that also appear in some comparisons of differentiated samples are presented in other figures). Unexpectedly, low levels of *UCP1* expression were also detected in some preadipocyte samples, and this was more pronounced in SC preadipocytes.

The DN and SC hASCs isolated from the nine independent donors were differentiated and analyzed by laser-scanning cytometry [6,7,26,27]. The efficiency of their differentiation to adipocytes defined by expression of general fat cell markers did not differ significantly by the site of origin or differentiation protocol (Figure 2A). The most prominent adipogenic marker genes [37] (Appendix A) were expressed significantly higher (except *STAT3*) in all mature adipocyte samples as compared to progenitors and were not significantly different between SC and DN adipocytes or samples differentiated with white and brown protocols. Hierarchical cluster analysis of these genes confirmed that the samples clustered into two main groups according to their differentiation status but not by tissue origin or the differentiation protocol used, suggesting that these factors had no effect on the general adipocyte differentiation efficiency of the progenitor cells (Figure 2B).

### 3.2. Differentiated Adipocytes from DN Progenitors Display Higher Browning-Related Gene-Expression Features

Next, we investigated morphology and differential transcriptomic characteristics of adipocytes differentiated from hASCs of the nine donors. Laser-scanning cytometric data clearly showed that the DN adipocyte population, differentiated by either the white or the brown protocol, had more brown cells characterized by the appearance of UCP1 and small lipid droplets (Figure 2C). To compare tissue origin and differentiation protocol-dependent transcriptional changes related to browning, recently developed computational tools ProFAT and BATLAS were applied. When the ProFAT browning gene-set [22] was used to score browning probability, browning markers were expressed higher in the DN samples irrespective of the differentiation protocol (Figure 2D,E) and appeared in distinct clusters (Appendix A). Expression of BATLAS genes visualized as heat map by hierarchical clustering (Figure 2F) also showed that the differentiated DN and SC samples predominantly appeared in these two main distinct clusters. Interestingly, even the white differentiation protocol resulted in BATLAS defined brown cell content (Figure 2G) while the highest brown cell percentage appeared in DN samples with noticeable donor variance (which was apparent in ProFAT scores as well). Six out of the BATLAS browning marker genes (*UCP1*, *CPT1B*, *EHHADH*, *AKAP1*, *SOD2*, and *ACSL5*) were expressed significantly higher in differentiated DN samples (Appendix A). Interestingly, the classical brown adipocyte marker *ZIC1* [16], was lowly expressed in both DN and SC adipocytes, and its expression was significantly increased only when the SC progenitor samples were differentiated by the brown protocol; these data questions the applicability of *ZIC1* as a lineage-specific, brown adipocyte marker gene.

Checking the expression of UCP1 at the protein level in differentiated adipocytes, we detected the highest UCP1 content in brown adipocyte samples of DN origin (Figure 2H), while in adipocytes of SC origin, the UCP1 expression was lower. Based on the above cytometric, gene expression, and metabolic data, it can be concluded that adipocytes differentiated from DN progenitors have greater thermogenic potential as compared to those originated from SC adipose tissue, suggesting that cultivated hASC of distinct anatomical locations maintain their different potentials for browning ex vivo.

### 3.3. Differential Gene Expression Patterns Reveal Network Pathways Associated with Higher Browning Potential of DN Adipocytes

After establishing the higher browning potential of cell populations derived from DN, we examined the pattern of DEGs between the DN and SC samples. Considering both the white and the brown differentiation protocol, 1049 genes were expressed differentially in matured DN adipocytes. Approximately half of these genes were already differentially expressed at a preadipocyte state (272 were higher and 250 lower expressed in DN), but the other half became expressed differentially only in differentiated adipocytes (257 were higher and 270 lower expressed in DN adipocytes) (Figure 3A,B). Relating the DEGs in mature adipocytes to how they were expressed in preadipocytes, four distinct groups could be defined. Figure 3A shows a global heat map representation of the expression profile in the four groups.

257 genes (Group 1) were expressed at a greater extent in differentiated DN adipocytes as compared to SC ones and this differential expression was not observed in DN versus SC preadipocytes (Figure 1B,C). This group includes well-established browning marker genes such as *UCP1*, *PRDM16*, *CKMT1A/B*, *CIDEA*, and *PM20D*1 (Appendix A). In further analyses, interaction relationships among products of higher expressed genes could be revealed using the STRING computational tool, which also defined enriched pathways (KEGG and REACTOM), biological processes, or cell compartments (Figure 3C,D, and Appendix A by Gephi). Most of the browning marker genes mentioned above appeared in a separate cluster (black square). Interestingly, signaling pathways, such as interferon alpha/beta, rhodopsin-like receptors, the TRAF3-dependent IRF activation pathway, and cytokine-cytokine receptor interactions, were predominant among DN specific clusters, whereas metabolic pathways were scarce. These significantly enriched pathways were clearly linked to differentiated DN brown cells, as they did not appear in DN preadipocytes as enriched pathways when compared to SCs (Figure 1B–D). We then sought to find linkers among the characteristic functional modules of the DN specific interaction network, which might have the most pronounced effect on the structure of the entire interactome. We calculated the betweenness centrality scores to point out important genes that connect different functional modules and the number of bridges to learn how many modules are connected by this gene. This network analysis confirmed the key role of *UCP1* to maintain network integrity. Other genes, such as *ESR1*, *MT2A*, *LEPR2*, *IRF7*, and *AGT*, may also be pivotal for this network (Table 1, Group 1).

In total, 272 genes (Group 2) were expressed at a greater extent in DN as compared to SC preadipocytes, and their expression remained elevated in DN adipocytes as compared to SC ones after white or brown differentiation, presumably contributing to a higher propensity to browning (Figure 3A,B). The most affected pathways were complement and coagulation cascades, signaling by retinoid acid and interaction between L1 and ankyrins (Figure 3E,F, and Appendix A by Gephi). As expected, pathways in this group were already revealed in the analysis of higher expressed genes in DN versus SC preadipocytes (Figure 1B–D; Italic font: pathway also appears in preadipocyte state; Capital letters: Pathway also appears in FTO obesity-risk allele-based comparison shown below). Similarly, expression of ECM and lipid metabolism genes is different in progenitors and this difference remains significant in mature adipocytes. Additionally, transcriptional regulation by Homeodomain/Homeobox containing transcriptional factors was tissue-specific and not affected by the differentiation state. The *BMP4* gene seems to be of paramount importance for a stable network structure; its role was already established for adipogenesis, promotion of adipocyte differentiation, and brown fat development, where it induces UCP1 expression [38]. In addition, *CD34*, *EFNA5*, *RARB*, and *ALDH1A3* genes appeared, according to the betweenness and bridges scores, to have importance in this interactome network (Table 1, Group 2). *TBX1*, a well-established beige marker gene [17], also appeared in this group as an important network member.

### 3.4. Downregulated Pathways in DN Adipocytes

In total, 270 genes (Group 3) were downregulated in differentiated DN adipocytes compared to SCs and this differential expression was not seen in the preadipocyte state; expression of some of these genes may be non-permissive for browning. STRING network and pathway analysis showed that these genes were predominantly involved in remodeling of the ECM, cell adhesion, and synthesis of GPI-anchored proteins pathways. Cytokine signaling was also highlighted as several interleukin receptors (e.g., *IL1RE*, *IL18RE*, *IL20RE*), *TGFB3*, and *KIT* were expressed higher in the matured SC adipocytes (Figure 4A,B, and Appendix A by Gephi). *COL1A1* plays a central role in the network, but *GPC3*, *KRT7*, *CNR1*, and *SEMA3A* are also important in sustaining a stable network structure (Table 2, Group 3).

250 genes (Group 4) were expressed at a lower extent in DN preadipocytes as compared to SC and their expression remained low in the DN adipocytes as compared to SC ones after white or brown differentiation; these may be part of pathways which are not permissive for browning either. ECM organization pathways had outstanding dominance among the related pathways and the repeated appearance of the GPI-anchored protein pathway (SC preadipocytes and Group 3) emphasizes the importance of glycosylphosphatidylinositol-anchored membrane proteins in the formation of the SC phenotype. In addition, the transcription regulation by homeobox and homeodomain and the neuroactive ligand-receptor interaction pathways also appeared in this group (Figure 4C,D, and Appendix A by Gephi). When analyzing relationships, *POSTN* was the most significant, but *RUNX2*, *NCAM1*, *GATA2*, *EDIL3*, and *PAX3* could also be important for maintaining network connectivity. Based on their network position, the *IRX1*, *IRX3*, and *GRIA3*, expressed higher in SC adipocytes, may have roles in restraining browning potential (Table 2, Group 4).

### 3.5. Shared PPARγ Mediated Gene Expression Patterns in DN and SC Adipocytes

Both ProFAT and BATLAS estimations clearly showed that SC precursor cells also reacted to brown as compared to the white differentiation protocol by upregulating brown genes. The degree of upregulation of these genes could be similar in DN and SC adipocytes and missed when DN and SC data were compared. Therefore, we analyzed genes that responded similarly to brown differentiation protocol, that is, long-term rosiglitazone exposure in SC and DN adipocytes. Indeed, out of the 217 genes responding by upregulation to brown differentiation in either DN or SC adipocytes, only 40 (Appendix A) were also present in Groups 1 and 2 (529 genes) described above. Eighty genes were expressed higher in brown adipocytes, irrespective to their anatomical origin, and they displayed enrichment of the PPAR signaling pathway (Figure 5A–C). Regarding the latter glycerol kinase, *PCK1*, *CPT1B* were present among BATLAS genes [21], *PLIN5* was linked to browning [39], *ANGPTL4* had not been investigated yet in this respect. In an earlier report, Loft et al. claimed that browning of human adipocytes, induced by PPARγ stimulation by rosiglitazone of stem cells isolated from the prepubic fat pad of a 4-month-old male donor, required the metabolic regulator Kruppel-like factor 11 (KLF11) and reprogramming of a PPARγ super-enhancer [40]. We found that *KLF11* was among the rosiglitazone upregulated genes (Figure 5D) in either SC or DN adipocytes suggesting that it may be involved in the regulation of browning of neck area adipocytes.

Among the upregulated browning genes outside of the PPAR pathway, *CIDEA* was listed by ProFAT and also found in the Group 1 genes above, similar to *PM20D1*, which was strongly influenced by natural genetic variations in humans [41] and *CPT1B* which was essential for beta-oxidation of fatty acids. Gamma-butyrobetaine dioxygenase (BBOX1) catalyzes the formation of L-carnitine from gamma-butyrobetaine and has been reported to play a role in adipocyte browning [42]. The beige marker *CITED1* is a transcriptional coactivator of the p300/CBP-mediated transcription complex [25]. Higher expression of glycerol phosphate acyltransferase (GPAT3) in SC and DN brown adipocytes reflects increased triacylglycerol synthesis capacity. The mitochondrial membrane transporter for long-chain fatty acids (SLC27A3) was also upregulated in response to PPARγ stimulation. There were several other highly upregulated genes (e.g., *OAS*, *CD96*, *KRT79*), which should be investigated in the future for their possible role in brown adipocyte function. Altogether, these data show that in addition to DN anatomical location, browning potential is also influenced by the appearance of PPARγ ligands in both SC and DN adipocytes.

The browning protocol also downregulated a large number of genes in either DN or SC preadipocytes; additionally, 35 overlap with Groups 3 and 4 (Appendix A). Out of the downregulated genes, 70 were downregulated in cells from both DN and SC origin with the indication of the importance of glutathione and arachidonic acid metabolism pathways (Figure 5E,F).

### 3.6. Presence of the FTO Obesity-Risk Alleles Significantly Influences Browning Gene Expression Profiles

Donor selection was based on the presence of obesity-risk SNP of the FTO locus, so donors 1 to 3, 4 to 6, and 7 to 9 carried a T/T risk-free, a T/C heterozygous, or a C/C obesity-risk genotype, respectively. ProFAT analysis of the expression profile of white and brown marker genes revealed that 11 brown marker genes were significantly lower (*ACAA2*, *SLC25A20*, *ACO2*, *HADHB*, *ETFDH*, *SDHB*, *NDUFS1*, *ACADS*, *DLD*, *ETFB*, *PDHA1*) and two white markers (*ALCAM*, *HOXC8*) were significantly higher expressed in FTO C/C obesity-risk genotype (Figure 6A, red rectangle). Evaluation by BATLAS further highlighted FTO obesity-risk allele dependent differences, as 23 out of these genes were significantly lower expressed in samples carrying the FTO C/C obesity-risk allele (Figure 6B, red rectangle). Hierarchical cluster analyses based on Pearson correlation showed that 91% of the FTO T/T donor samples clustered together and 83% of the FTO C/C obesity-risk donor samples appeared in the same main cluster and they were closer to the preadipocytes, considering ProFAT genes (Appendix A). BATLAS genes assembled very similarly, 75% of the FTO T/T and 83% of the FTO C/C donor samples clustered in the same main groups and FTO obesity-risk samples appeared closer to the non-thermogenic preadipocytes. Heterozygous samples appeared in either the FTO T/T or the C/C group (Appendix A). The ProFAT and BATLAS data clearly suggest that hASCs, which carry the FTO C/C obesity-risk genotype, have significantly lower browning potential.

As a next step, we investigated all the DEGs based on the presence of the FTO obesity-risk allele in the white- and brown-differentiated DN and SC samples, and 1295 DEGs were identified. The expression of 624 and 671 genes was significantly lower or higher, respectively, in the adipocytes that carried the risk alleles as compared to the non-risk allele carriers. The expression level of these genes was not as much influenced by the FTO status in the preadipocyte state suggesting that these genes have a functional role in the matured adipocytes (Figure 6C). Analysis of these 1295 DEGs revealed that 81 code mitochondrial proteins, according to the Human MitoCarta 2.0 database [43] (Appendix A). Out of those, 16 belong to one of the mitochondrial complexes and were lowly expressed in the C/C obesity-risk genotype. Only six out of the 81 mitochondrial protein-encoding genes showed higher expression in the obesity-risk genotype (*SEPT4*, *PYCR1*, *PRELID2*, *CPT1A*, *MTHFD1L*, and *FKBP10*). The expression profile of genes encoding mitochondrial proteins (1035 genes in our samples, out of the 1158 genes in MitoCarta 2.0 database) visualized by a heat map (Appendix A) also demonstrated that the majority of the mitochondrial genes were expressed at a lower extent in samples that carried the obesity-risk genotype. The genes *CKMT1A* and *CKMT1B*, which encode key mitochondrial creatine kinases and reported to be involved in UCP1-independent thermogenesis [4], were among those being significantly less expressed in FTO obesity-risk samples.

Among the genes that were expressed at lower levels in samples carrying the FTO obesity-risk allele, metabolic pathways were most affected including lipid metabolism, thermogenesis, carbon metabolism, oxidative phosphorylation, and degradation of certain amino acids such as valine, leucine and isoleucine (Figure 6D,E). There was a striking resemblance between the set of network pathways negatively affected by the C/C genotype and those which were revealed by analyzing the ProFAT and BATLAS gene products (Figure 6H–K). Furthermore, out of the 23 significantly enriched KEGG pathways defined by ProFAT genes, 21 were also found in BATLAS, and 19 of these are defined by genes that showed suppressed expression in FTO C/C samples. Out of the 29 KEGG pathways defined by BATLAS genes, 25 were also found in pathways determined by the lower expressed genes in C/C samples. After analyzing the Reactome pathways, similarly, high agreements could be found (Appendix A). Interactome network analysis of the lower expressed genes in FTO obesity-risk genotype suggests that under normal conditions, *DECR1* (dienoyl-CoA reductase), *LIPE*, *LDHB*, *SOD2*, *ATP5B,* and *J* are central elements in maintaining connectivity (Appendix A). Genes that were linked to adipo- and thermogenesis, such as *SLC2A4*, *EHHADH*, *PPARGC1A*, *CPT1B*, and *FABP4*, appeared with high betweenness scores. In addition to the metabolic genes, a cytokine ligand, *CCL2*, and a receptor, *CXCR4*, also appeared as important critical components of this interactome network.

Comparing the FTO status-related gene expression patterns to upregulated DEGs in DN versus SC adipocytes, we found that out of the 529 genes (Groups 1 and 2) that were significantly higher in the DN samples, 33 genes were poorly expressed in the FTO obesity-risk genotype samples (Figure 6L and Appendix A). Pathway analysis of genes with different expression based on tissue origin and presence of the FTO obesity-risk allele revealed that different pathways dominated the two. Interferon signaling and G-protein coupled rhodopsin-like receptors were the prominent DN specific pathways. The FTO C/C alleles influenced metabolic pathways, e.g., fatty acid and carbon metabolism. The 33 genes which had high expression in thermogenically more potent DN and appeared downregulated in the FTO obesity-risk genotype adipocytes underline the importance of PPAR signaling, lipid metabolism pathways (*CPT1B*, *FABP3*, *EHHADH*, *PLA2G4A*, *ZDHHC19*), creatine kinase activity (*CKMT1A* and *CKMT1B*), carboxylic acid metabolic processes (*HYAL1*), and sodium channel regulator activity (*SGK2*, *GPLD1*) in determining browning potential.

Since the presence of the FTO risk allele may lead to loss of restrain on the expression of genes related to thermogenesis and its regulation, we also looked for and found DEGs that showed higher expression in obesity-risk genotype samples (Figure 6F,G). Clearly, the expression of genes related to the organization of the ECM was the most prominent upregulated pathway. This is consistent with the finding that ECM organization pathway has high importance in restraining the thermogenic potential (see gene expression and enrichment analyses in Groups 3 and 4 above and in preadipocytes). In addition, integrin cell surface interaction, TGFB, PI3K-Akt, and ras signaling appeared among significantly enriched pathways in FTO obesity-risk samples. The network analysis clearly indicated the importance of fibronectin (*FN1* gene) for the integrity of this group of networks, but also the position and relationship of others such as *CTGF*, *BDNF*, *TGFB*, and *GLI1* (Appendix A); these growth and transcription factors regulate cell proliferation, differentiation, and growth and can modulate the expression of other growth factors to determine cell fate. Considering the relationship between the DN downregulated genes and those expressed higher in C/C alleles, among the 520 genes (Groups 3 and 4) whose expression was lower in DN samples, 71 were higher in FTO obesity-risk samples (Figure 6L and Appendix A). Again, the importance of ECM organization and, in particular, the collagen degradation and integrin cell surface interactions (*COL1A1*, *COL8A1*, *COL8A2*, *COL13A1*, *SDC1*, *ACAN*, *HAPLN1*, *ITGA11*) came into focus. After further analyses of these shared gene-sets, overexpressed pro-inflammatory cytokines and cytokine receptors (e.g., *IL11*, *IL20RA*, *IL27RA*) could be linked to less thermogenic DN adipocytes, and thereby, to obesity and inflammation [44]. Finally, *IRX3*, which was already described as a negative [19] and positive [45] regulator of the development of thermogenic adipocytes, was also among the genes kept lowly expressed in DN and FTO T/T adipocytes, but became higher expressed in the adipocytes with the obesity-risk genotype.

## 4. Discussion

Human ASCs from distinct fat depots have the potential to undergo a browning program [14]. One of the major sites that contain active BAT in humans is located in the neck, particularly in its deep regions [14,46]. When we compared SC and DN adipocyte progenitors from human neck, the latter had greater browning potential, in accordance with previous findings [18,36,47,48]. On the other hand, SC hASCs were also able to build up a significant thermogenic competency. The difference between SC and DN hASCs in the capability of building a thermogenic potential might be explained by their differential depot-specific epigenetic chromatin signatures. Divoux et al. reported that subcutaneous preadipocytes cultivated from abdominal and gluteofemoral fat of six overweight or obese women had depot-specific chromatin structures resulting in gene expression patterns that were selective for the anatomical identity of the cells [49]. Primarily, we found 1420 DEGs when the hASCs of SC and DN origins were compared. This relatively large number of DEGs and their designated pathways suggest that the two anatomically localized tissues have cells of different epigenetic signatures even at precursor level and presumably have a specific ECM structure that is maintained ex vivo after differentiation. The organization of the ECM is differently regulated: degradation processes are prominent in the SC samples, whereas in DN samples, ECM reorganization and regulation of vascular development appear to be significant. The retinoic acid signaling pathway also appeared in DN cells both at progenitor and at differentiated states; its role in inducing thermogenesis via increasing angiogenesis by activating VEGFA/VEGFR2 signaling has been suggested but not explored in detail [50,51].

Many genes were already proposed to predispose cells to a higher or lower thermogenic potential at the precursor level [48,52,53,54]. Similar to these results, in our samples, *EBF2*, *LHX8*, *MEOX2*, and *TBX15* showed higher expression in DN preadipocytes, while *HOXC8*, *HOXC9*, *DTP*, and *IGFBP3* were pronounced in SC ones. Several white and brown predisposing marker genes identified by Xue et al. [48] were also differentially expressed in our SC and DN preadipocytes, such as *S1PR3*, *GPRC5A*, *MASP1*, *C10orf90*, *ST6GALNAC3*, and *SVIL* having higher expression in DN, whereas *COL12A1*, *SHROOM3*, *GRIK2*, and *HAPLN1* were being enriched in SC precursors. However, the 26 white and 25 brown predisposing marker genes pointed out by this group are highly diverse in their function and no determining pathways can be identified from the gene set that would underpin future thermogenic activity [48].

Recently, two global approaches based on mouse and human experimental data have identified sets of genes (50 in ProFAT and 119 in BATLAS), the expression pattern of which could be used, also by us, to determine the browning potential of adipocyte cells or tissues [21,22]. Surprisingly, the number of common genes in the two lists is very small, only 17. However, we found that ProFAT and BATLAS gene products are actually part of very similar pathways linked to increased metabolic activity as defined by KEGG or REACTOM, suggesting that it may be more appropriate to focus on gene-defined pathways, biological processes, or specific cell components for better defining adipocyte browning.

In this study, we considered three different influences (anatomical origin, PPARγ stimulation by rosiglitazone, and alleles of the rs1421085 FTO locus) asking what pathways and processes they induce for increased browning. Regarding site-specific gene expression, we can conclude that among the 1049 DEGs in DN versus SC samples, several classical brown and beige markers are greatly co-expressed in mature DN adipocytes, e.g., *PRDM16*, *LHX8*, *EBF2*, *TBX1*, *KCNK3*, *CITED1*, and *MTUS1*. The brown markers *LHX8* [16], *PRDM16* [12,55], and *EBF2* [52,53] also showed significantly higher expression in the DN samples, while the beige marker *TBX1* [17] had a similar pattern. Interestingly, several well-known thermogenic markers, including *ELOVL3* [56] and *PPARGC1A* [57], were not expressed differently in SC and DN adipocytes. A few brown marker genes had higher expression in SC samples differentiated by white protocol (*DIO2* [1], *HSPB7*, *EVA1A* [58]) and *ZIC1* [16] was barely expressed in all samples. Our findings are in line with the conclusions of Jespersen et al. [18], who claimed that there is an overlap signature between classical murine brown and beige marker genes in adipocytes in the DN region of adult humans.

Among the pathways identified by 1049 DEGs comparing SC and DN samples, the differential regulation of ECM organization is a characteristic feature of adipocytes of the two origins, suggesting site-specific organization of the ECM, which is maintained even when the cells are differentiated ex vivo. This points out that conditions used in the cell culture, such as the quality of plates, application of biomaterials, or scaffold-based three-dimensional culture technics, may significantly influence the ECM organization, and consequently, cellular functions. Additionally, this suggests that efforts should be made to use culture methods that are the most consistent with in vivo conditions. In addition, signaling transmitted by class A1 rhodopsin-like receptors and interferon alpha/beta receptors, which were expressed higher in DN samples, and neuroactive ligand receptors, ras proximate 1 (RAP1) G-protein, and cGMP molecules, which appeared elevated in SC samples, may be critical in establishing of the different functional properties of DN and SC adipocytes. The significantly enriched interferon, cytokine, and other signaling pathways in DN adipocytes suggest that the anatomical origin of the progenitors determine extrinsic factors involved in cell-cell communication rather than the metabolic properties of the browning adipocytes. There are several secreted factors which were already reported to positively influence browning in an autocrine manner [59].

Alternative sources of mesenchymal stem cells (MSCs), e.g., the oral derived MSCs, are able to release immunomodulatory and trophic molecules, which support the regeneration of damaged tissues [60]. In the future, these less invasively harvestable MSCs, which have great potential in regenerative medicine, might be used to conduct studies with regard to adipocyte browning [61]. However, our data indicate that the tissue origin of the progenitors may significantly affect the results of the studies, which could give uncertainty in experiments using induced pluripotent stem cells (iPSC) or other non-adipose-derived MSCs [61,62]. It also suggests that it is preferable to use cells obtained from tissues with high thermogenic potential in future brown adipocyte transplantation and regenerative assays. MSCs isolated from various tissues have distinct differentiation potential and usually pathways that reflect the original tissue are preferred.

We found 217 genes that were upregulated either in SC or DN adipocytes in response to the brown protocol with sustained rosiglitazone treatment. Eighty of these genes (and the PPAR signaling pathway) were significantly upregulated in adipocytes of both origins, including *KLF11*, *TSHR*, *PDK4*, *APOL6*, *CPT1B*, and *CIDEA*, which showed elevated expression in response to rosiglitazone in browning hMADS-derived adipocytes as well as a result of the formation of PPARγ super-enhancers [40]. Our data suggest that KLF11-dependent super-enhancers are assembled upon rosiglitazone treatment in both types of primary adipocytes from the neck.

Another super-enhancer, which is located on an obesity-risk associated locus of the intronic region of the FTO gene, was linked to the regulation of browning in subcutaneous fat [19,45,63,64] by determining the level of IRX3 and IRX5. The role of IRX3 and IRX5 in this signaling pathway is still controversial [19,45,63]. In our study, *IRX3* was significantly higher expressed in samples with FTO obesity-risk alleles and support its suppressive role in thermogenesis. Interestingly, DN adipocytes (progenitors and differentiated) had also lower expression of IRX family members, including *IRX1-3* and *5-6* as compared to SC adipocytes, irrespective to their FTO allele status, which suggests that an FTO rs1421085 SNP-independent mechanism in DN samples suppresses IRX gene expression. Overall, the expression profile and pathway analysis of the 1295 DEGs related to the absence and presence of the C/C alleles, clearly points to major difference in regulation of metabolism including mitochondrial biological processes. Out of these, 624 genes (e.g., *CKMT1A/B*, *CITED1*, *PPARGC1A/B*, and *CPT1B*) were lower expressed in those adipocytes that carried the obesity-risk genotype and the enriched pathways (maintained by the normal T/T genotype) included fatty acid metabolism, thermogenesis, respiratory electron transport, and the signaling by retinoic acid (which showed DN specific appearance as well). Surprisingly, the enriched pathways identified by these gene expression patterns include almost entirely the pathways defined by the BATLAS and ProFAT brown marker genes (Appendix A), which suggest that this genetic predisposition has a pivotal importance in determining thermogenic competency of adipocytes from the neck. In addition to the regulation of metabolic pathways, the organization of the ECM and various signaling pathways (RAP1, Ras, Hippo, Relaxin, TGF-beta, PI3-Akt) appears to be greatly affected by the risk alleles. In spite of the existing 104 similarly expressed DEGs in the comparisons of FTO T/T vs. C/C genotypes and DN vs. SC adipocytes (Appendix A), FTO genotypes define largely different gene expression patterns and pathways compared to those recognized on the basis of tissue origin.

Comparing the results of the three different analytic approach used for characterization of gene expression patterns linked to browning (by anatomical origin, PPARγ stimulation and presence of the rs1421085 FTO locus), only five transcripts (carnitine palmitoyltransferase 1B, creatine kinase mitochondrial 1B, thiamine transporter 2, family with sequence similarity 189 Member A2, and long intergenic non-protein coding RNA 2458) were expressed higher in brown adipocytes determined by all of the three conditions (Appendix A). Only one gene, Collagen alpha-1(VIII) chain, was significantly higher expressed in those samples that showed lower thermogenic activity. The low number of common genes and pathways suggests that these three factors elicit significantly different responses in cells in contribution to the browning transcriptome and phenotype. Interestingly, the tissue origin resulted in about two times (2.2×) more DEGs, presumably associated with browning, than PPARγ stimulation. It is striking that the absence or presence of the FTO obesity-risk allele results in the greatest difference in the gene expression profile (1295 genes) of the differentiated cells (Appendix A). Consequently, research efforts targeting the regulatory system determined by the rs1421085 FTO locus have the potential to develop novel therapeutic approaches for increasing weight loss by thermogenesis in obese patients.

UCP1-independent energy releasing processes were recently described in adipocytes, which may provide an alternative way to reduce obesity. Our RNA sequencing data showed that *CKMT1A* and *B*, futile cycle maintaining creatine kinases [4], were expressed higher in DN as compared to SC adipocytes and less abundant in those that had an obesity-risk genotype of the rs1421085 FTO locus. *PM20D1* encodes an enzyme that catalyzes the synthesis of N-lipidated/N-fatty-acyl amino acids, which function as endogenous uncouplers of mitochondrial respiration in a UCP1-independent manner [6]; it was expressed higher in DN brown adipocytes and rosiglitazone increased its expression in both SC and DN adipocytes regardless of the FTO allele status. This specific effect of the PPARγ-agonist was observed in other studies as well [9,41].

We found several genes with enriched expression in DN as compared to SC adipocytes, which have not been directly linked to browning thus far. The product of relaxin receptor 2 (RXFP2) mediates G-protein dependent stimulation of adenylate cyclase and an increase of cAMP levels [65]. Receptor-type tyrosine-protein phosphatase N2 (PTPRN2) was shown to act as an autoantigen in type I diabetes and be required for the accumulation of insulin-containing vesicles, preventing their degradation in rat gastrointestinal endocrine cells [66]. Moreover, it has a role in the accumulation of norepinephrine, dopamine, and serotonin in autonomic nerve endings [67]. Ephrin type-A receptor 5 (EPHA5), which has a role in regulation of insulin secretion [68], was also highly expressed in DN as compared to SC brown adipocytes. Cysteinyl leukotriene receptor 2 (CYSLTR2) was shown to induce type 2 immune response in an interleukin (IL)-33 dependent manner [69]; this response was proposed, but also debated to be a positive regulator of browning in mice [70,71]. In our experiments, transcripts of both genes were enriched in DN adipocytes. Further investigations are needed to confirm the direct role of these genes in the browning process.

## Figures and Tables

**Figure 1 cells-09-00987-f001:**
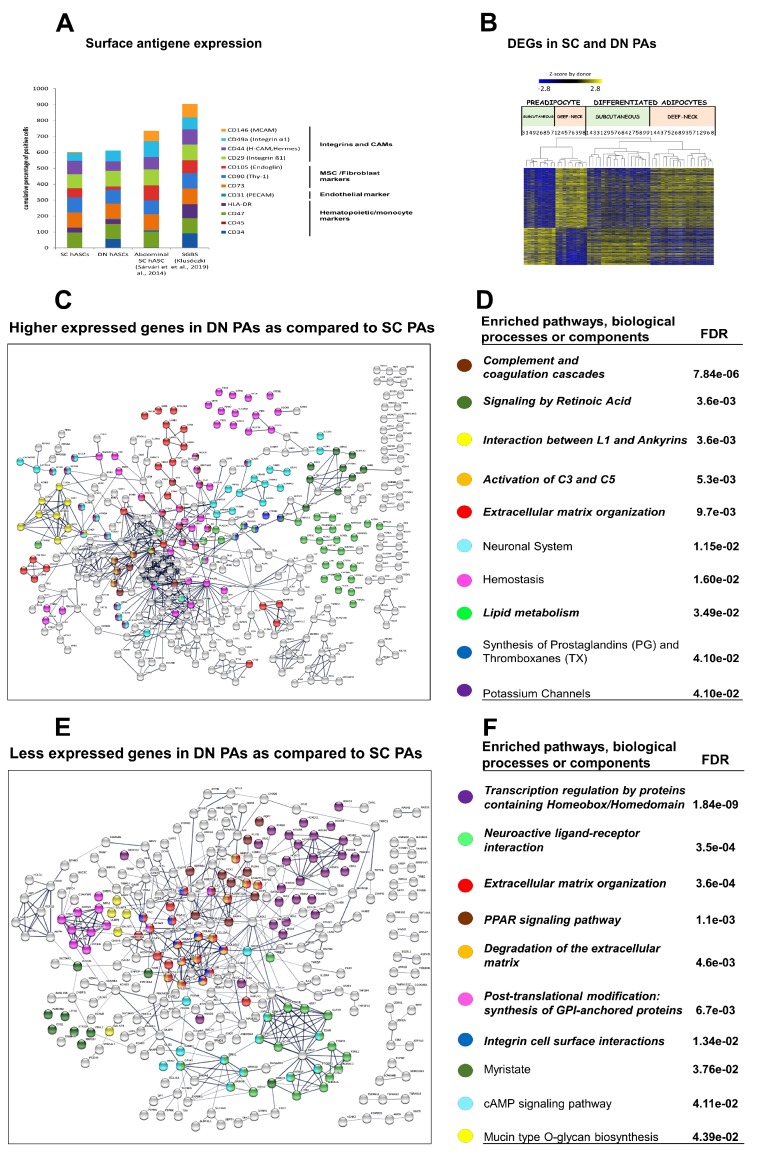
Characterization of subcutaneous (SC) and deep-neck (DN) progenitors. (**A**) Expression of 17 surface markers was determined in preadipocytes of three independent donors with T/C genotype for the rs1421085 locus in the FTO gene by flow cytometry. The numbers represent the percentage of positive cells. *n* = 3 (**B**) Heat map visualization of 1420 differentially expressed genes (DEGs) in preadipocytes based on tissue origin; 1 to 9 labels the individual from which the sample was obtained. (**C**,**E**) Interactome maps of genes greater expressed in DN preadipocytes (**C**) and SC preadipocytes (**E**). Edges represent protein-protein based interactions; interaction score confidence value: 0.7. (**D**,**F**) Selected significant Enriched pathways, cell components or biological processes based on DN vs. SC DEGs in preadipocytes with false discovery rate (FDR) values; Italic font: pathway also appears after differentiation (**D**) higher expressed in DN (**F**) and in SC preadipocytes; *n* = 9 donors, six samples/donor.

**Figure 2 cells-09-00987-f002:**
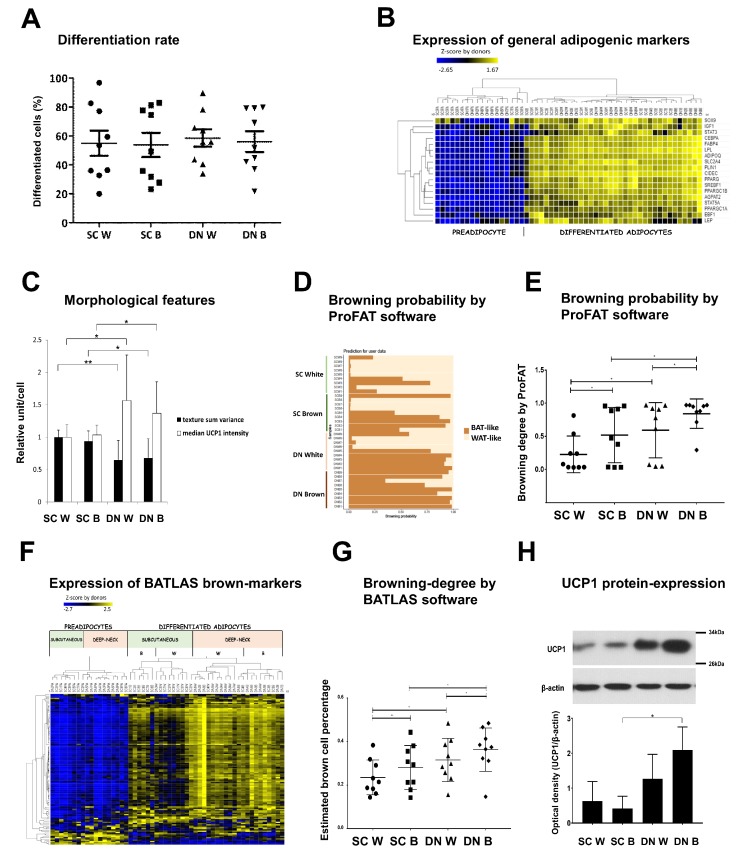
The degree of differentiation and browning of subcutaneous (SC) and deep-neck (DN) adipocytes. (**A**) Ratio of differentiated adipocytes over the total number of cells. (**B**) Heat map displaying the expression profiles of general adipocyte marker genes. (**C**) Texture sum variance and median Ucp1 protein content of adipocytes (as compared to SC white adipocytes) per cell. In every experiment, 2000 cells per each sample were recorded and measured. (**D**) Based on the raw gene-expression data the browning probability of the samples was analyzed and visualized by ProFAT software. (**E**) Browning probability considering tissue origin and differentiation protocol. (**F**) Expression profile of BATLAS marker genes in SC and DN adipose progenitors and differentiated samples. (**G**) Estimation of brown cell percentage by BATLAS considering tissue origin and differentiation protocol, *n* = 9 donor, four samples/donor. (**H**) Expression of UCP1 in adipocyte lysates of one representative donor detected by immunoblotting. Densitometry of immunoblots was performed with samples of three independent donors. W: white differentiation protocol; B: brown differentiation protocol. 1 to 9 labels the individual from which the sample was obtained. In the comparison of two groups, paired two-tailed Student’s t-test, in multi-factor comparison, two-way ANOVA and post hoc Tukey’s test was used, * *p* < 0.05, ** *p* < 0.01.

**Figure 3 cells-09-00987-f003:**
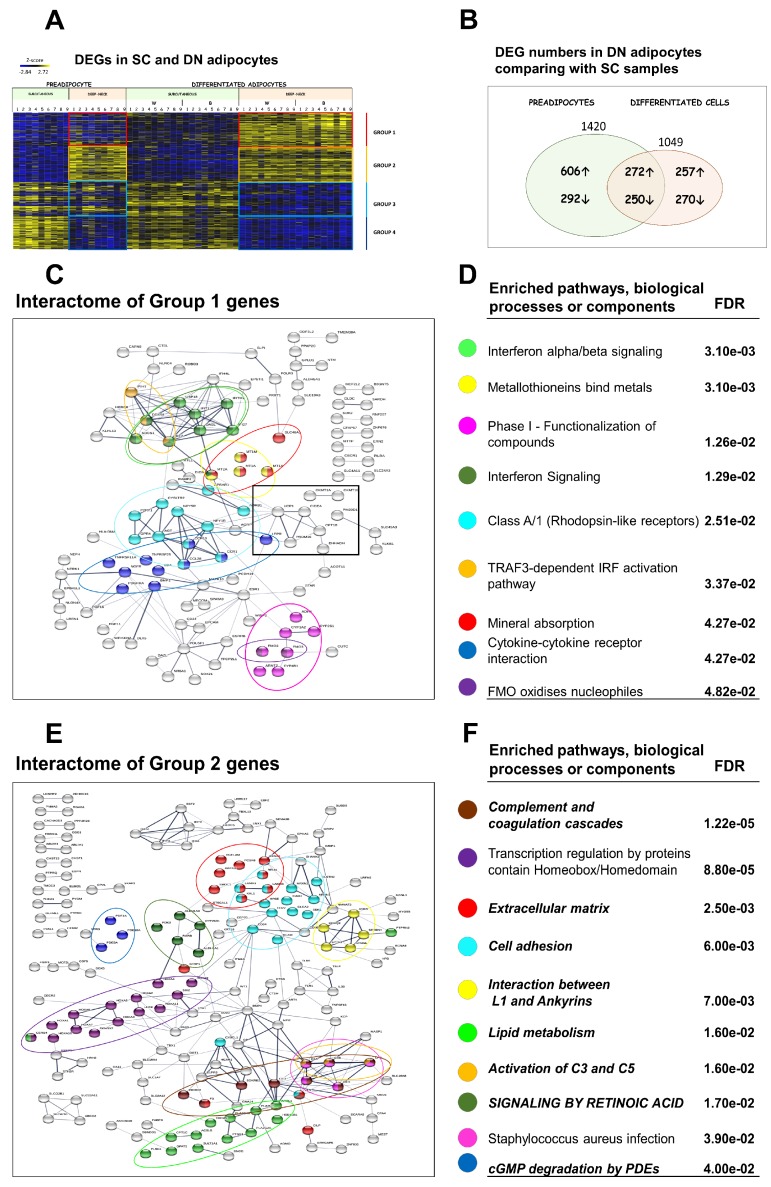
Differentially expressed genes (DEGs) in subcutaneous (SC) and deep-neck (DN) mature adipocytes. (**A**) Heat map shows the expression profile of 1049 DEGs in SC and DN mature adipocytes. Based on how these DEGs are expressed in preadipocytes, four distinct groups were formed (Group 1–4). 1 to 9 labels the individual from which the sample was obtained. (**B**) Venn diagram shows the DEG numbers based on DN and SC comparison at preadipocyte and differentiated adipocyte state. (**C**) Interactome map and (**D**) Selected Significantly Enriched pathways and their false discovery rate (FDR) value based on DN vs. SC DEGs in Group 1; interaction score confidence value: 0.4. (**E**) Interactome map and (**F**) Selected significantly Enriched pathways and their FDR values in Group 2; interaction score confidence value: 0.4. Edges represent protein-protein based interactions. Italic font: pathway also appears in preadipocyte state; Capital letters: Pathway also appears in FTO obesity-risk allele-based comparison. W: white differentiation protocol; B: brown differentiation protocol.

**Figure 4 cells-09-00987-f004:**
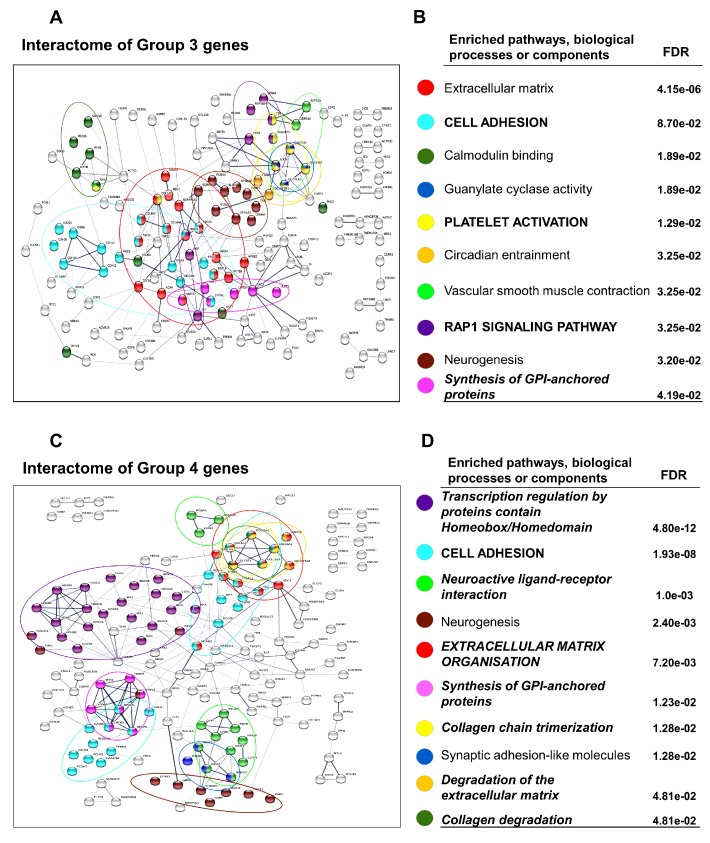
Interactome maps and enriched pathways, biological processes, or cellular components expressed lower in deep-neck (DN) as compared to subcutaneous (SC) adipocytes. Groups of genes were defined in Figure 3A. (**A**) Interactome map and (**B**) Selected Significantly Enriched pathways and their false discovery rate (FDR) values based on DN vs. SC differentially expressed genes in Group 3; interaction score confidence value: 0.4. (**C**) Interactome map and (**D**) Selected significantly Enriched pathways and their FDR values in Group 4; interaction score confidence value: 0.4. Edges represent protein-protein based interactions. Italic font: pathway also appears in preadipocyte state; Capital letters: Pathway also appears in FTO obesity-risk allele-based comparison.

**Figure 5 cells-09-00987-f005:**
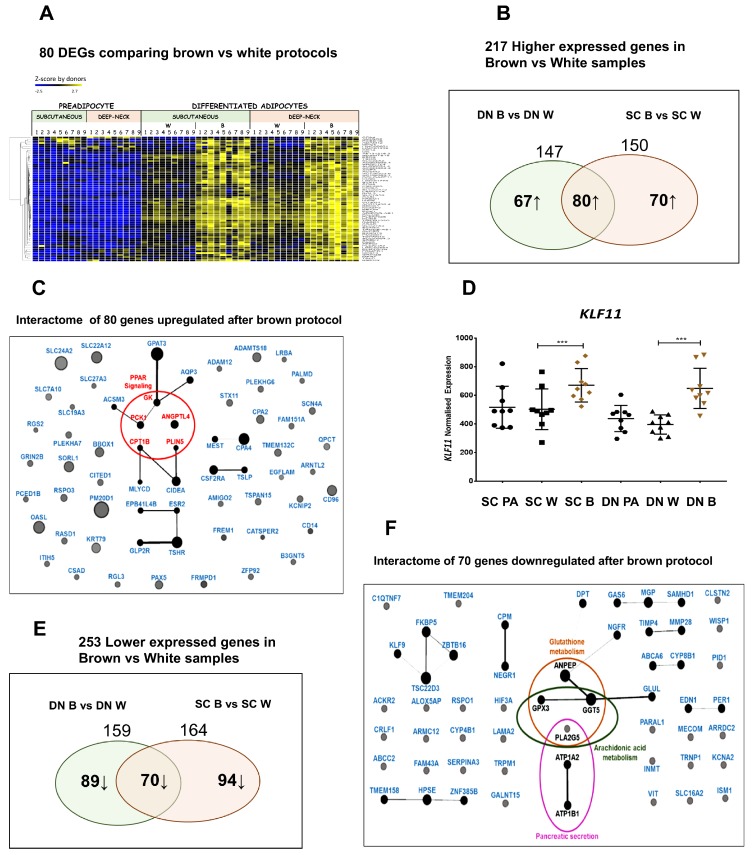
Expression of genes up- and downregulated after browning protocol in both subcutaneous (SC) and deep-neck (DN) adipocytes. (**A**) Heat map of the upregulated genes. 1 to 9 labels the individual from which the sample was obtained. (**B**,**E**) Venn diagrams of the number of genes responding to brown differentiation. (**C**,**F**) Representation of overlapping up- and downregulated genes, respectively, in a Gephi interactome map where the size of circles is proportional to the fold change in expression values. (**D**) Expression profile of *KLF11* based on DESeq normalized RNAseq data, statistics: GLM *** *p* < 0.001. PA: preadipocytes, W: white differentiation protocol, B: brown differentiation protocol.

**Figure 6 cells-09-00987-f006:**
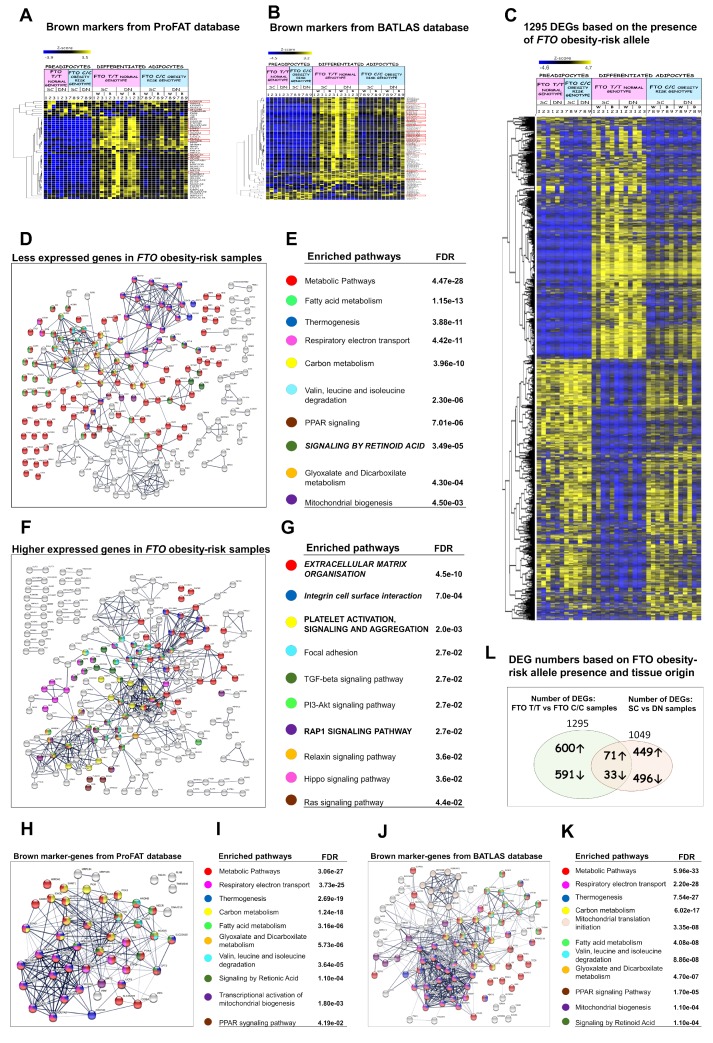
Gene expression analyses based on the presence of FTO obesity-risk allele. (**A**) Heat map shows the expression profile of marker genes from ProFAT database. (**B**) Heat map visualization shows differences of brown marker genes based on BATLAS database; samples ordered; *n* = 6. (**C**) Heat map shows the expression profile of differentially expressed genes (DEGs) based on the presence of FTO obesity-risk alleles in SC and DN adipose progenitors and differentiated samples. (**D**,**E**) Interactome map and selected significantly enriched pathways, biological processes or cell components and their false discovery rate (FDR) values of lower expressed genes in FTO obesity-risk allele carrier samples, interaction score confidence value: 0.7. (**F**,**G**). Interactome map and selected significantly enriched pathways, biological processes or cell components and their FDR values of higher expressed genes in FTO obesity-risk allele carrier samples. Edges represent protein-protein based interactions. SC: subcutaneous DN: deep-neck, W: white differentiation protocol; B: brown differentiation protocol. (**H**,**I**) Interactome map and Significantly enriched pathways with their FDR values of ProFAT browning marker genes; interaction score confidence value: 0.4. (**J**,**K**) Interactome map and Significantly enriched pathways with their FDR values of BATLAS browning marker genes; interaction score confidence value: 0.4. Italic font: pathway also appears in preadipocyte state; Capital letters: Pathway also appears in FTO obesity-risk allele based comparison. (**L**) FTO genotype affected genes among the DEGs in DN and SC samples. 1 to 9 labels the individual from which the sample was obtained.

**Table 1 cells-09-00987-t001:** Betweenness centrality (BC) score and Bridges number identified based on STRING Interactome network analysis by Igrapg R and Fold Changes (FC) identified by DESeq in R of the differentially expressed genes DN vs. SC comparison. Average FC when two data were considered (Groups 1 and 2 respectively), *n* = 9 donors.

Group	Gene	BC Score	Bridges	FC	Gene Description
1	*ESR1*	1445	3	2.0	estrogen receptor 1
	*UCP1*	1323	1	6.1	uncoupling protein 1 (mitochondrial, proton carrier)
	*MT2A*	1061	2	2.4	metallothionein 2A
	*LEPR*	988	1	1.9	leptin receptor
	*IFIH1*	957	0	2.0	interferon induced with helicase C domain 1
	*IRF7*	880	2	2.0	interferon regulatory factor 7
	*NLRC4*	864	1	1.9	NLR family, CARD domain containing 4
	*CTSL*	813	1	1.9	cathepsin L
	*AGT*	745	2	2.4	angiotensinogen (serpin peptidase inhibitor, clade A, member 8)
	*MAPK10*	720	2	1.8	mitogen-activated protein kinase 10
2	*BMP4*	2776	8	3.1	bone morphogenetic protein 4
	*CD34*	2048	3	3.6	stem cell adhesion
	*EFNA5*	1519	1	2.2	ephrin-A5
	*RARB*	1362	0	3.9	retinoic acid receptor, beta
	*ALDH1A3*	1194	1	4.4	aldehyde dehydrogenase 1 family, member A3
	*SMOC1*	1138	1	3.2	SPARC related modular calcium binding 1
	*TBX1*	1125	3	2.5	T-box 1
	*RELN*	1087	0	3.1	reelin
	*SEMA3B*	1050	1	2.2	immunoglobulin domain (semaphorin) short basic domain, secreted, 3B
	*SPTA1*	1046	1	2.1	spectrin, alpha, erythrocytic 1 (elliptocytosis 2)

**Table 2 cells-09-00987-t002:** Betweenness centrality (BC) score and Bridges number identified based on STRING Interactome network analysis by Igrapg R and Fold Changes (FC) identified by DESeq R of the differentially expressed genes DN vs. SC comparison. Average FC when two data were considered (Groups 3 and 4 respectively), *n* = 9 donors.

Group	Gene	BC Score	Bridges	FC	Gene Description
3	*COL1A1*	2057	4	2.1	collagen, type I, alpha 1
	*GPC3*	1113	1	3.0	glypican 3
	*KRT7*	914	2	2.9	keratin 7
	*CNR1*	864	1	5.8	cannabinoid receptor 1 (brain)
	*THY1*	822	2	2.0	Thy-1 cell surface antigen
	*SEMA3A*	781	5	2.8	immunoglobulin domain, secreted (semaphorin) 3A
	*FBN2*	630	2	6.4	fibrillin 2
	*ITGA4*	617	3	2.7	integrin (antigen CD49D) alpha 4 subunit of VLA-4 receptor
	*KCNJ6*	569	0	4.2	potassium inwardly-rectifying channel, subfamily J, member 6
	*KIT*	543	3	7.0	v-kit Hardy-Zuckerman 4 feline sarcoma viral oncogene homolog
4	*POSTN*	1541	5	5.8	periostin, osteoblast specific factor
	*RUNX2*	1454	5	2.9	runt-related transcription factor 2
	*NCAM1*	1230	4	3.4	neural cell adhesion molecule 1
	*GATA2*	1050	3	3.5	GATA binding protein 2
	*EDIL3*	901	3	4.6	EGF-like repeats and discoidin I-like domains 3
	*PAX3*	840	4	87.1	paired box 3
	*IRX1*	768	3	23.4	iroquois homeobox 1
	*NTNG1*	760	1	7.0	netrin G1
	*HAND2*	752	2	30.5	heart and neural crest derivatives expressed 2
	*ITGB2*	663	2	2.6	integrin, beta 2 (complement component 3 receptor

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
