# Peer review of "FTO Intronic SNP Strongly Influences Human Neck Adipocyte Browning Determined by Tissue and PPARγ Specific Regulation: A Transcriptome Analysis"

_cells, 2020, doi:10.3390/cells9040987_

Round 1

Reviewer 1 Report

This manuscript is a collection of differentially expressed genes (DEGs) in primary adipocytes isolated from human between deep-neck area (DN) and subcutaneous-neck area (SC).  The authors indicated that adipocytes from DN expressed highly brown-adipogenic gene and lower extracellular matrix genes compare to adipose cells from SC, and these differences were related with PPAR gamma.  In addition, fat mass and obesity-associated (FTO) gene SNP was related with low gene expression of adipo-browning factor (i.e. thermogenic). These findings are interesting.  I recommend the paper to be published in this journal with some revisions. 

The main weakness of this manuscript is having some unclear points.  Main novelty of this paper is regional difference of gene expression profiles on human pre-adipocytes isolated from DN and SC area by the surgical method.  Why did these cells keep browing-phenotypes (i.e. activation by PPAR gamma) separated from bodies and without blood flow or/and nerve stimulus.  So, it is my consideration that this manuscript is publishable after additional discussion about this question.

This experience needs a patient's consent.  This point should be write clearly in methods part.

Author Response

Response to Reviewer Comments

Reviewer 1

This manuscript is a collection of differentially expressed genes (DEGs) in primary adipocytes isolated from human between deep-neck area (DN) and subcutaneous-neck area (SC).  The authors indicated that adipocytes from DN expressed highly brown-adipogenic gene and lower extracellular matrix genes compare to adipose cells from SC, and these differences were related with PPAR gamma.  In addition, fat mass and obesity-associated (FTO) gene SNP was related with low gene expression of adipo-browning factor (i.e. thermogenic). These findings are interesting.  I recommend the paper to be published in this journal with some revisions.

We thank the Reviewer for the positive comments and useful suggestions which helped to improve the quality of the manuscript.

The main weakness of this manuscript is having some unclear points.  Main novelty of this paper is regional difference of gene expression profiles on human pre-adipocytes isolated from DN and SC area by the surgical method.  Why did these cells keep browning-phenotypes (i.e. activation by PPAR gamma) separated from bodies and without blood flow or/and nerve stimulus.  So, it is my consideration that this manuscript is publishable after additional discussion about this question.

Answer: The reason why deep neck precursors keep their browning capability even in ex vivo conditions might be that they have a depot-specific chromatin profile that result in higher expression of genes which facilitate their commitment to thermogenic adipocytes.  To reveal this, specific experiments might be planned in future studies (Discussion was modified between lanes 669-677; reference 49 was added).

This experience needs a patient's consent.  This point should be write clearly in methods part.

Answer: Written informed consent was obtained from all participants before the surgical procedure (Included to lanes 124-127).

Reviewer 2 Report

The topic of this article entitled “ FTO intronic SNP strongly influences human neck adipocyte browning determined by tissue and PPARγ specific regulation: a transcriptome analysis” is interesting and within the journal's scope. Nevertheless, this reviewer would suggest some improvements, before further considerations. The study has certainly new information related to the role of FTO obesity-risk allele.

The main strength of this paper is related to the appropriateness of approach and to the study design: really relevant, as well as the soundness of the whole article and the relevancy of discussion. The clarity of writing is also fine; however, some key-concepts should be slightly improved for a major clarity to the readers, involving other critical discussion and comparing better some evidence with the previously published papers.

Authors have investigated on some types of MSCs “The ratio of brown and white adipocytes is partially determined during the early differentiation of mesenchymal progenitors into adipocyte subtypes which is strongly influenced by genetic predisposition [19,20]” however, poor has been discussed on other MSCs that could be easily harvested and well-described in the literature, such as the oral-derived stem cells, which can support the immunomodulatory activity in the local environment (Please, see and discuss: Potential use of human periapical cyst-mesenchymal stem cells (hpcy-mscs) as a novel stem cell source for regenerative medicine applications. Front Cell Dev Biol 2017, 5, 103.)

Authors have consistently discussed on the role of ECM, and they clearly reported “Among the pathways identified by 1049 DEGs comparing SC and DN samples, the differential regulation of ECM organization is characteristic feature of adipocytes of the two origins, suggesting site specific organization of the ECM, which is maintained even when the cells are differentiated ex vivo.” However, none about the role of “biomaterials” or “scaffolds” as study model has been reported. On the other hand, it’s interesting to discuss something about the novel approaches with nanotechnologies.

Finally, authors reported in their work the hypothetical role of MSC, especially hASCs in the sentence: “Our results suggest that cultivated hASCs from distinct locations of the human neck still keep their differing propensity for adipocyte browning which is strongly influenced by the presence of the obesity-risk alleles at the FTO intronic locus.” however, to date, it’s well-known the concern related to use of protocols potentially tumorigenic. In this light it’s important to briefly describe on safe in-vitro reparative models, working without any additive (e.g. BSA) to apply safely such protocols in human models.

Author Response

Response to Reviewer Comments

Reviewer 2

The topic of this article entitled “ FTO intronic SNP strongly influences human neck adipocyte browning determined by tissue and PPARγ specific regulation: a transcriptome analysis” is interesting and within the journal's scope. Nevertheless, this reviewer would suggest some improvements, before further considerations. The study has certainly new information related to the role of FTO obesity-risk allele.

The main strength of this paper is related to the appropriateness of approach and to the study design: really relevant, as well as the soundness of the whole article and the relevancy of discussion. The clarity of writing is also fine; however, some key-concepts should be slightly improved for a major clarity to the readers, involving other critical discussion and comparing better some evidence with the previously published papers.

We are grateful to the Reviewer for her/his insightful remarks which helped us to interpret our data more correctly.

Authors have investigated on some types of MSCs “The ratio of brown and white adipocytes is partially determined during the early differentiation of mesenchymal progenitors into adipocyte subtypes which is strongly influenced by genetic predisposition [19,20]” however, poor has been discussed on other MSCs that could be easily harvested and well-described in the literature, such as the oral-derived stem cells, which can support the immunomodulatory activity in the local environment (Please, see and discuss: Potential use of human periapical cyst-mesenchymal stem cells (hpcy-mscs) as a novel stem cell source for regenerative medicine applications. Front Cell Dev Biol 2017, 5, 103.)

Answer: The major goal of our study was to find key molecular regulators of human adipocyte browning by understanding which factors determine the capability of hASC populations to the commitment for thermogenesis. Our data suggest that genetic, depot-specific and differentiation-related factors together influence the browning potential of the progenitors. One possible explanation might be that these influences determine epigenetic chromatin signatures resulting in gene expression patterns that are selective for the particular populations of the stem cells. It does not exclude the possibility that easily harvestable stem cells (e.g. oral derived MSCs) or iPSC can be used to conduct studies with regard to adipocyte browning, and in the future they might be applicable to generate implantable anti-obesity heat-producing adipocytes. However, if possible, it is preferable to use cells obtained from tissues with high thermogenic potential for these studies. Adipose tissue is relatively easy to obtain and contains 200 times more stem cells than the equal volume of bone marrow, and we think it is a good candidate in the future for large scale manufacturing processes as well (Discussion was modified between lanes 741-750; references 60-62 were added).

Authors have consistently discussed on the role of ECM, and they clearly reported “Among the pathways identified by 1049 DEGs comparing SC and DN samples, the differential regulation of ECM organization is characteristic feature of adipocytes of the two origins, suggesting site specific organization of the ECM, which is maintained even when the cells are differentiated ex vivo.” However, none about the role of “biomaterials” or “scaffolds” as study model has been reported. On the other hand, it’s interesting to discuss something about the novel approaches with nanotechnologies.

Answer: Differential regulation of ECM organization is a characteristic feature of progenitors of SC and DN origins, which is maintained even when the cells are differentiated ex vivo. Application of biomaterials or scaffold-based 3D cell culture technics may significantly influence the ECM organization and consequently cellular functions and suggests that efforts should be made to use these methods that are the most consistent with in vivo conditions. However, the method how to use these biomaterials and scaffolds in different applications (micropatterned structures, spheroid technology or 3D bioprinting) alters the viability and differentiation potential of the cells, which has to be investigated in the future (Discussion was modified between lanes 729-732).

Finally, authors reported in their work the hypothetical role of MSC, especially hASCs in the sentence: “Our results suggest that cultivated hASCs from distinct locations of the human neck still keep their differing propensity for adipocyte browning which is strongly influenced by the presence of the obesity-risk alleles at the FTO intronic locus.” however, to date, it’s well-known the concern related to use of protocols potentially tumorigenic. In this light it’s important to briefly describe on safe in-vitro reparative models, working without any additive (e.g. BSA) to apply safely such protocols in human models.

Answer: In this study, we applied previously optimized adipogenic differentiation protocols to develop white or brown adipocytes ex vivo. The differentiation was carried out in serum and additive-free DMEM-F12-HAM medium (added to lane 132) that contained some of the following compounds: apo-transferrin, insulin, T3, dexamethasone, hydrocortisone, rosiglitazone and IBMX. Interestingly, we found that the FTO rs1421085 allele status and the anatomical origin of the progenitors had a greater effect on their browning potential than the difference in the obtained protocols. Of note, hASCs were cultivated in 10% FBS containing medium before the differentiation process. It is known, that the FDA guidelines let us to use bovine-derived products especially in cellular therapeutic products. In fact, Mendicino et al. (Cell Stem Cell, 2014) reviewed all MSC regulatory filings and found that over 80% of all regulatory submissions described the use of fetal bovine serum (FBS) during the hMSC manufacturing process. Several other analyses of hMSC-based clinical trials in the recent years have similarly shown that at least 65-75% of the trials utilize FBS (Ikebe and Suzuki, Biomed Res Int, 2014; Minonzio and Linetsky, CellR4, 2014). In future studies aiming to generate transplantable brown adipocytes to fight against obesity, more safe protocols should be applied, whereas several alternatives for FBS, including human platelet lysate (HPL) and chemically defined, serum-free medium (SFM) and other more safe components could be used.

Round 2

Reviewer 2 Report

Suggestions have been fairly addressed. Paper could be considered for publication.